# Did COVID-19-Related Alcohol Sales Restrictions Reduce Alcohol Consumption? Findings from a National Online Survey in South Africa

**DOI:** 10.3390/ijerph19042422

**Published:** 2022-02-19

**Authors:** Marieke Theron, Rina Swart, Mukhethwa Londani, Charles Parry, Petal Petersen Williams, Nadine Harker

**Affiliations:** 1School of Public Health, Faculty of Community and Health Sciences, University of the Western Cape, Robert Sobukwe Road, Bellville 7535, South Africa; rswart@uwc.ac.za; 2Alcohol, Tobacco and Other Drug Research Unit, South African Medical Research Council, Francie van Zijl Drive, Cape Town 7505, South Africa; mukhethwa.londani@mrc.ac.za (M.L.); charles.parry@mrc.ac.za (C.P.); petal.petersen@mrc.ac.za (P.P.W.); nadine.harker@mrc.ac.za (N.H.); 3Department of Psychiatry, Stellenbosch University, Cape Town 7505, South Africa; 4Department of Psychiatry and Mental Health, University of Cape Town, Cape Town 7935, South Africa; 5School of Public Health and Family Medicine, University of Cape Town, Cape Town 7935, South Africa

**Keywords:** Facebook, illegal alcohol sales, COVID-19 pandemic, lockdown, heavy episodic drinking, coping mechanisms, anxiety, depression and alcohol policies

## Abstract

Background: South Africa has a high prevalence of heavy episodic drinking (HED). Due to the high levels of alcohol misuse and violence, public hospital intensive care units were often overrun during the COVID-19 pandemic. This research investigated alcohol intake behaviour change during differing levels of lockdown restrictions, which included bans on alcohol sales. Methods: A self-reported Facebook survey ran from July to November 2020. The questions included socio-demographics, income, alcohol intake, purchasing behaviour, and reasoning. Chi-square tests/Fisher’s exact test for categorical data, Student’s *t*-test for normal continuous data, and the Mann–Whitney U test for non-normal data were applied. Multiple logistic regression was run for HED versus moderate drinkers. Results: A total of 798 participants took part in the survey, of which 68.4% were female. Nearly 50% of participants fell into the HED category and the majority bought alcohol illegally during restrictions. HED respondents who drank more alcohol than usual during restrictions reported that they felt stressed, needed to relax, and were bored. Conclusions: Policies intended to increase the pricing of alcohol may have the potential to reduce alcohol intake. Reducing stress and anxiety may be key to curtailing HED during emergency situations.

## 1. Introduction

In 2020, the World Health Organization (WHO) declared the novel 2019-nCoV (coronavirus disease 2019) a global pandemic [1], with over 256 million infections and more than 5 million deaths worldwide to date, and numbers are still increasing [2].

In South Africa (SA), this led to the declaration of a national state of disaster by the government [3] and subsequent restrictions on various activities such as outdoor exercise, dog-walking, travel, and purchasing alcohol and cigarettes. These restrictions were imposed by amending the Disaster Management Act [4].

Widespread uncertainty and fear of the disease itself, in addition to secondary consequences such as home confinement and various restrictions, have had a severe impact on mental health globally [5]. Nearly 50% of South Africans reported symptoms of anxiety and depression in a study conducted from 20 to 31 May 2020 [6].

In some countries, alcohol was classified as an essential item, while in SA, the sale and transport of alcohol were banned (as gazetted in the Disaster Management Act) for a total of 161 days during four different periods from 27 March 2020 to date. These periods were 27 March to 1 June (63 days), 12 July to 17 August (35 days), 27 December 2020 to 1 February 2021 (35 days), and 27 June 2021 to 25 July 2021 (28 days) [7]. Whilst the ban on alcohol sales was necessary to increase hospital bed availability, illegal alcohol sales suddenly increased to supplement the population’s demand for alcohol, and a new age of prohibition started [8].

SA has the sixth highest per person alcohol intake in the world, accounting for 64.6 g of pure alcohol per drinker per day, and has a very high prevalence of heavy episodic drinking (HED) or excessive alcohol intake in those who consume alcohol [9,10,11]. HED is defined as drinking more than 60 g of absolute alcohol on any one occasion in the past month [12]. This translates to having five small glasses of wine or around one and a half shooters of spirits, and is usually anything more than six drinks during one occasion [10]. Statistics show that even though there are not as many consumers of alcohol in this country as in most European countries, those who do use alcohol consume more alcohol than the average person in other countries [9,12].

As soon as the alcohol ban was imposed in SA, there was a significant decrease in emergency hospitalisations and unnatural deaths, indicating the extent to which alcohol abuse leads to the need for emergency services and has severe health impacts that call for hospitalisation [13,14]. 

According to research on harmful alcohol use in SA, it costs the country 12% of GDP every year [15], which was equal to ZAR 135 billion in 2021 [16]. Alcohol is the fifth largest contributor to years of life lost in SA [9], adding to the total disease burden due to chronic diseases and conditions such as various cancers of the digestive system, blood, reproductive system, and breast; diabetes; mental and behavioural disorders, including depression and epilepsy; and lastly, cardiovascular-related diseases [17]. Data show that lower socio-economic groups in SA are disproportionately affected by these alcohol-related illnesses [18]. 

The WHO suggests specific interventions that countries may implement to reduce alcohol intake, such as improved monitoring and surveillance systems, increasing excise taxes, establishing minimum unit pricing, enforcing bans on alcohol advertising, restricting the physical availability of alcohol, reducing the density of retail outlets, providing psychosocial interventions for people with harmful alcohol intake, and lastly, providing prevention, treatment, and care for alcohol use disorders in health and social services [9]. The COVID-19 lockdown and alcohol sales restrictions provided a natural experiment to investigate both the effect that a physical reduction in alcohol had on alcohol intake patterns, and the rationales provided by both moderate and heavy episodic drinkers. 

This research aimed to investigate the socio-demographic profile of participants, their alcohol intake behavioural changes during differing levels of lockdown restrictions, and factors linked to HED during the initial months of the COVID-19 pandemic in SA.

## 2. Materials and Methods

### 2.1. Type of Study

An online ad-hoc self-reported survey ran from 28 July 2020 to 28 November 2020 using the social media platform Facebook. 

### 2.2. Population and Sample

Participants in the study were citizens of South Africa or permanent residents over the age of 18. The social media platform Facebook streamlined the questionnaire to ensure it would only be visible to people living in South Africa, and age was asked in the survey.

A convenience sample was used. A sample size of 310 was calculated [19] to find sufficient females with HED, as HED is usually less prevalent in females than in males [10]. Thus, a sample size of 798 was deemed to be adequate to use for analysis. 

### 2.3. Instrument and Procedure

This tool was developed by Massey University and adapted by the South African Medical Research Council in conjunction with the International Alcohol Control Study. The multi-national online survey was modified to be responsive to local circumstances (such as the national currency, ethnic groupings, and geographic sub-areas) (Table 1). The questions and optional answers that were used in this study are shown in Table 1.

### 2.4. Statistics

Data cleaning and checking were executed in Microsoft Excel (Microsoft corporation, Washington, USA) [20] prior to being imported into SPSS (IBM corporation, New York, USA) [21] and STATA (STATA corporation, Texas, USA) [22] software. Numerical data were checked for normality using the Shapiro–Wilk test. Normal data are reported as mean (standard deviation) and non-normal data as median (interquartile range), while categorical data are reported as frequencies and percentages. Data were analysed in an iterative way by first performing explorative analyses, then searching for associations using Chi-square tests/Fisher’s exact test for categorical data, Student’s *t*-test for normal data, and the Mann–Whitney U test for non-normal data. After identifying variables that were significantly associated with the dependent binary variable of HED and moderate drinking, these variables were entered to compile a standard multiple logistic regression in STATA [22]. The variables that were entered into the model to calculate the adjusted odds ratio (AOR) were sex, age, change in monthly net household income since the COVID-19 pandemic restrictions, aged 70+ or serious medical condition or immunocompromised, frequency of alcohol consumption, illegal alcohol purchasing, first time buying alcohol illegally, restrictions made it harder to cut down on drinking, and alcohol makes social/physical distancing more difficult. Multicollinearity was assessed by examining correlations between predictors. No two predictors had a correlation of more than 0.5. Model fit was checked using an adaptation of Hosmer Lemeshow’s goodness of fit test, and all models indicated appropriate fit. *p*-values less than 0.05 were considered statistically significant. Participants with missing data, *n* = 569 or 41.2%, were more likely to be of coloured ethnicity (Fisher’s exact, 8.742, *p* = 0.026). However, gender, age, province, and change in income did not differ significantly for missing data.

### 2.5. Ethical Considerations

Participants were anonymized and responses were stored in secure firewalled facilities that were password protected. Access was only given to research team members. Information was given upfront regarding alcohol addiction help lines and the contact details of researchers for more information. Participants were assured that participation was voluntary and were asked to indicate their agreement to participate by choosing ‘agree’ or ‘not agree’ to participate, which were used to indicate signatory agreement of the informed consent sheets. This research was given ethical approval by the South African Medical Research Council (ref: EC017-6/2020) and the University of the Western Cape Biomedical Research Ethics Council (ref: BM21/5/11).

## 3. Results

The final survey comprised 798 people (68.4% female). The majority of participants were between the ages of 55 and 64 (29.3%). A significantly younger cohort of females between 18 and 54 years of age completed the survey at a rate of 55% compared to males in this age group at 48% (*p* = 0.028; Chi-square test) (Table 2). The highest frequency of participants reported to be of white ethnicity, at 83%, and residing in Gauteng (40%) or the Western Cape (22%), and 65% of respondents reported a decrease in income during the pandemic.

The majority of participants reported that they did not suffer from any serious underlying medical conditions or were immunocompromised at the time of completing the survey. A significantly higher proportion of females than males completed the survey during stricter pandemic restrictions (*p* = 0.018). However, the majority of participants completed the survey during moderate pandemic restrictions, when people were required to stay at home, were allowed to interact with a few people outside of their household, some businesses and schools were open but public gatherings were banned, and physical distancing was required.

The majority of respondents reported to be alcohol consumers at 88.7%. In terms of drinking patterns (Table 3), males had a significantly higher frequency of daily alcohol consumption, while females were more likely to consume alcohol on a weekly or monthly basis (*p* = 0.008). A significantly higher frequency of males reported to binge drink on a daily and weekly basis (69.8% versus 51% for females) and more than 61.4% of males were classified as heavy episodic drinkers, compared to females at 42.7%. 

Significantly more males reported to have purchased alcohol illegally during the alcohol sales restrictions, while females who did buy alcohol illegally were significantly more likely to have done so for the first time (*p* = 0.039 and *p* = 0.002, respectively). When asked whether COVID-19 restrictions made it harder for participants who were trying to cut down on their drinking, significantly more females said that the restrictions made it harder to cut down (*p* = 0.031). The majority of participants, however, did not think that alcohol made social or physical distancing more difficult.

There was a significantly higher proportion of males in the heavy episodic drinking (HED) group compared to the moderate drinking group (Table 4), while there were significantly more females in the moderate drinking group. The HED group was significantly younger, with 66% of them below the age of 55.

There was no difference for the ethnicities, provinces, or restriction levels between the HED and moderate drinking groups.

Significantly more people in the HED group reported a reduction in their income after COVID-19 restrictions were introduced, purchased alcohol illegally, and reported that the pandemic restrictions made it more difficult for them to cut down on their alcohol consumption.

Moderate drinkers were significantly more likely to be older or have underlying medical conditions, reported that they consumed alcohol only monthly, that alcohol does make social distancing more difficult, and that if they had bought alcohol illegally, it was for the first time.

The HED group reported a significantly higher frequency of consuming more alcohol than usual than the moderate drinking group (*p* < 0.001) during both increased and decreased lockdown restrictions (Figure 1a,b). Similar frequencies were reported by heavy episodic drinkers and moderate drinkers of consuming the same amount of alcohol as usual during both increased and decreased restrictions. A significantly higher frequency of the moderate drinkers than drinkers in the HED group reported consuming less alcohol than usual during both increased and decreased lockdown restrictions. 

Reasons given for consuming more alcohol as the restrictions increased and decreased that were significantly more frequent for the HED group than for the moderate drinking group were, in descending order, feeling stressed, helping them to relax, feeling bored, having more time to consume alcohol with their household, and not needing to wake up for work/study (Figure 2 and Figure 3). Having more online social occasions was found to have a significantly higher frequency as the restrictions increased for HED only (Figure 2). Celebrating the lifting of restrictions and being able to see and drink with their friends was found to have a significantly higher frequency as the restrictions decreased for HED only (Figure 3).

Reasons given for consuming less alcohol as restrictions increased and decreased that were significantly more frequent for the moderate drinking group than for the HED group were that ‘it was more difficult to get alcohol while restrictions were placed on going out and while the shops were closed’ and that they had ‘not been able to socialise or go out or visit a pub’ (Figure 4a,b). Feeling that the restriction period was a good time to reduce how much they drank was found to have a significantly higher frequency as the restrictions increased for moderate drinkers only.

Multiple logistic regression analysis showed that compared to males, females had significantly lower odds of HED (AOR 0.30; 95% CI 0.13, 0.70; *p* = 0.006) (Table 5). Compared to 18–34-year-olds, people who were 65 and older had significantly lower odds of HED (AOR 0.23; 95%CI 0.06–0.91; *p* = 0.037). Compared to people who consumed alcohol every day or more, those who consumed alcohol 1–4 times per month had significantly lower odds of HED (AOR 0.13; 95%CI 0.03-0.45; *p* = 0.001).

Furthermore, compared to people who reported purchasing alcohol illegally during the pandemic alcohol sales restrictions, those who did not purchase alcohol illegally had significantly lower odds of HED (AOR 0.20; 95% CI 0.10–0.40; *p* < 0.001). While not statistically significant, people who stated that it was not difficult to cut down on their alcohol consumption, if they had already been trying to cut down when alcohol restrictions were implemented, also had lower odds of HED (AOR 0.49; 95% CI 0.24–1.00; *p* = 0.051).

## 4. Discussion

Lockdown restrictions may impede the efforts of those drinking harmfully to reduce their alcohol consumption. Heavy episodic drinkers reported drinking more alcohol than usual during both increased and decreased restrictions, and reported drinking more alcohol because they felt stressed, felt a need to relax, and felt bored. Significantly fewer people categorised as heavy episodic drinkers thought that the alcohol restrictions were a good incentive to reduce their alcohol intake. Nearly half of those surveyed fell in the HED category and the majority reported buying alcohol illegally during bans on alcohol sales.

This research found that the odds of being classified as a heavy episodic drinker were increased for people younger than 65, males, people who drink more frequently than monthly, people who bought alcohol illegally during the alcohol sales restrictions, and those who reported that reducing drinking was more difficult during the restrictions.

On 27 March 2020, South Africa implemented the first alcohol sales restrictions in response to the COVID-19 pandemic. The first sales ban lasted for about two months, the second and third lasted a month each, and the fourth lasted 28 days. These were the first alcohol sales restrictions in South Africa in 26 years and were implemented without forewarning, preventing the public from buying alcohol in advance. This research aimed to describe changes in drinking patterns and reasons given for these changes, specifically investigating the disparities between heavy episodic drinkers and moderate drinkers. 

Our study found that heavy episodic drinkers were prone to consuming more alcohol during restrictions, while moderate drinkers drank the same as usual, or less. Nearly half of the 798 participants who completed the Facebook survey were classified as heavy episodic drinkers, with more than 60% of males and 43% of females falling in the HED category. This is similar to a recent study conducted in Tshwane, South Africa, which reported 53% of adults as being heavy episodic drinkers [23]. These statistics may even be under-representative of the female HED prevalence due to the fact that the cut-off of more than six alcoholic drinks was used for both males and females in this questionnaire, whereas some alcohol intake studies use a cut-off threshold of four standard alcoholic drinks per occasion to identify HED in females [24]. 

People who were moderate drinkers reported drinking less or similar amounts of alcohol during both increased and decreased lockdown restrictions, while a significantly higher frequency of people with HED reported consuming more alcohol during both increased and decreased lockdown restrictions. This finding is similar to that of Meyers et al. (2021) [25], who found that participants taking part in a study in the Western Cape (*n* = 61) had one less day of heavy drinking during the alcohol restrictions. However, when they did drink alcohol again, the amount of alcohol consumed increased by an additional three units of alcohol per occasion. We found that 76% of heavy episodic drinkers bought alcohol illegally, showing that the majority of participants in this study found a way to buy alcohol, even though it was illegal. 

Research in the United States of America (USA) using a pre-pandemic and continuing pandemic survey showed that people who experienced higher levels of stress due to COVID-19 used alcohol to cope and drank alcohol more frequently [26]. Similarly, our study found that people reported drinking more alcohol due to feeling stressed or anxious, and the reasons for feeling stressed found in other SA studies were the lockdown regulations, alcohol and tobacco sales restrictions, the limitation of people’s freedom of movement, loss of jobs, and uncertainty about the future [27]. 

Reasons given in our study by heavy episodic drinkers for drinking more alcohol during increased restrictions that were significantly more frequent than for moderate drinkers were ‘feeling stressed/anxious’, ‘helps me relax/switch off’, and ‘I have been bored’. Research conducted two months before our Facebook survey in May of 2020, investigating the impact of the lockdown restrictions on the mental health of people, found that 46% and 47% of participants in the Western Cape achieved the diagnostic threshold for anxiety and depression, respectively (*n* = 860) [6]. Keeping the results of De Man et al. (2021) [6] and the results found in the Facebook survey in mind, people suffering with HED may be more prone to consuming greater amounts of alcohol during mentally challenging periods such as the emergency lockdown measures of 2020, due to feeling stressed and anxious. 

Increased lockdown restrictions were especially relevant to increased stress and anxiety compared to decreased lockdown restrictions, when wanting to relax was mentioned before stress. During decreased restrictions, the reasons heavy episodic drinkers gave for drinking more alcohol that were significantly more frequent than for moderate drinkers were ‘wanting to relax/switch off’, ‘feeling stressed out’, ‘alcohol shops were now open’, and ‘celebrating when COVID-19 restrictions were lifted/relaxed’. Interestingly, we see that ‘being bored’ moved from third to fifth most frequently mentioned.

Reasons given for drinking less alcohol were similar during increased and decreased lockdown restrictions. Significantly more moderate drinkers stated that they drank less alcohol because ‘it is more difficult to get alcohol with restrictions on going out and shops being closed’ and ‘I haven’t been able to socialise or go out to the pub’. Importantly, ‘money or cost’ reasons, the third most frequent, did not differ significantly between HED and moderate drinkers. The Facebook study found that 70% of heavy episodic drinkers experienced a reduction in their income compared to 60% of moderate drinkers. A report by the National Income Dynamics Study (NIDS) and Coronavirus Rapid Mobile Survey 2020 (CRAM) found that there was a 40% decline in active employment as well as a 10% decline in average earnings between February and April of 2020, three months before our Facebook survey [28]. 

Moderate drinkers stated that the ‘restrictions were a good time to reduce alcohol intake’ significantly more frequently than the HED group during increased restrictions; it was the fourth most frequent reason given for drinking less. 

Our research shows that limiting alcohol sales or imposing alcohol sales restrictions are not proven to be effective in reducing alcohol intake in people who are classified as suffering with HED. However, moderate alcohol consumers may benefit from these restrictions by using them as incentive to reduce their alcohol intake. This finding is similar to research by Meyers et al. (2021) [25]. In our research, we found that ‘money or cost’ was given as the third most frequent reason why both HED and moderate drinkers consumed less alcohol during the lockdown restrictions, indicating that a minimum unit price on alcohol or an increase in the price of alcohol may be a valid method of reducing the amount of alcohol that people consume [29].

A major limitation of this study was the fact that the majority (83%) of participants who completed the online Facebook survey were of white ethnicity and were older in age, and only two of the nine provinces were well represented, namely Gauteng and the Western Cape. The disparity in participation between ethnicities and provinces could be a result of the unequal accessibility to the Internet and electronic devices, as reported by Statistics South Africa (STATSSA), which showed that 70% of urban households had access to the Internet in 2017, while only 43% of rural households had Internet access [30]. Therefore, the findings are probably not generalisable to all ethnic groups in South Africa or to persons living in all provinces. Additionally, while we had every intention to ensure that participants were over 18, we were not able to verify/guarantee that this was indeed the case.

## 5. Conclusions

This research highlighted that people who are classified as heavy episodic drinkers react differently to the effects of emergency situations and concomitant policies that induce anxiety and stress. It was also found that policies intended to increase the pricing of alcohol, such as the WHO strategy of increasing excise taxes and minimum unit pricing, may have the potential to reduce alcohol intake in a time of crisis. 

Future research and policy interventions should investigate the effect that social support grants, food security measures, and other policies that improve living conditions may have on anxiety and alcohol intake as broad prevention strategies for chronic conditions related to alcohol. Additionally, support and treatment strategies should be given priority in the case of people with harmful levels of alcohol intake, as suggested by the WHO global strategy and health professionals around the world. 

## Figures and Tables

**Figure 1 ijerph-19-02422-f001:**
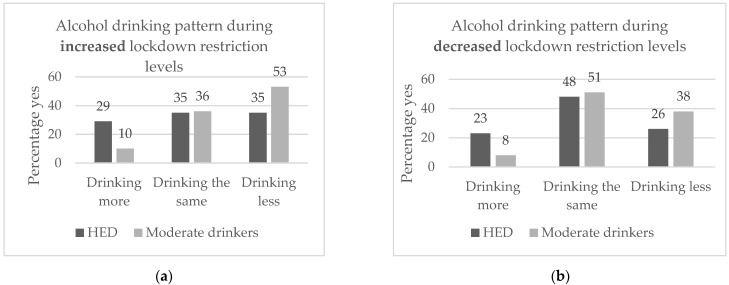
Alcohol drinking patterns of HED and moderate drinkers during (**a**) increased lockdown restriction levels (*p* < 0.001, Chi-square) and (**b**) decreased lockdown restriction levels (*p* < 0.001, Chi-square).

**Figure 2 ijerph-19-02422-f002:**
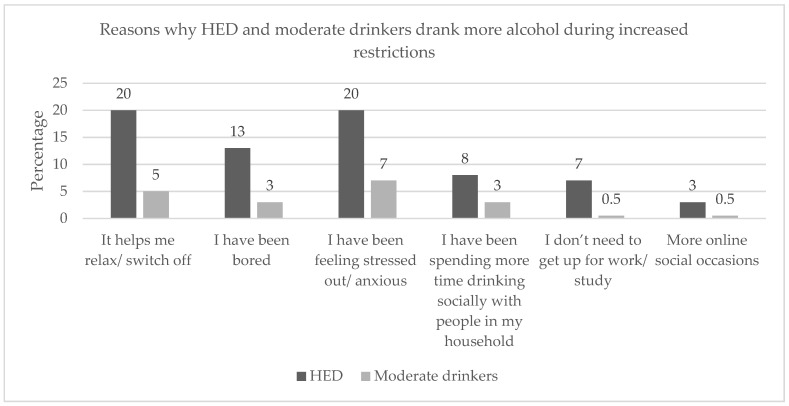
Reasons why HED and moderate drinkers reported drinking more alcohol during increased restrictions (all showed significant differences between HED and moderate drinkers; *p* < 0.05).

**Figure 3 ijerph-19-02422-f003:**
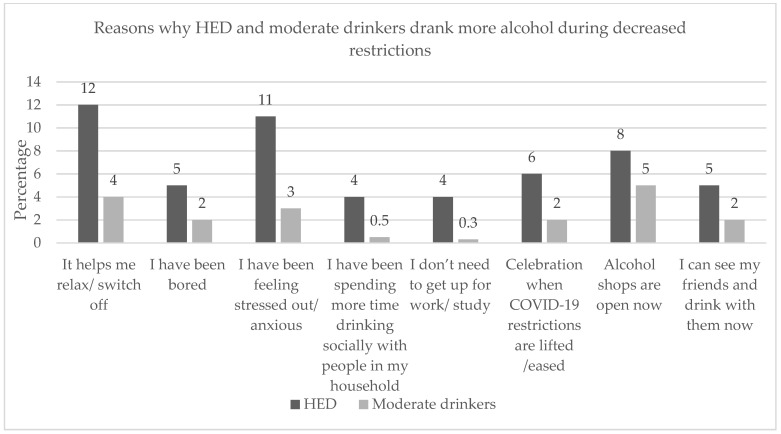
Reasons why HED and moderate drinkers reported drinking more alcohol during decreased restrictions (all except for ‘Alcohol shops are open now’ were significantly different between HED and moderate drinkers; *p* < 0.05).

**Figure 4 ijerph-19-02422-f004:**
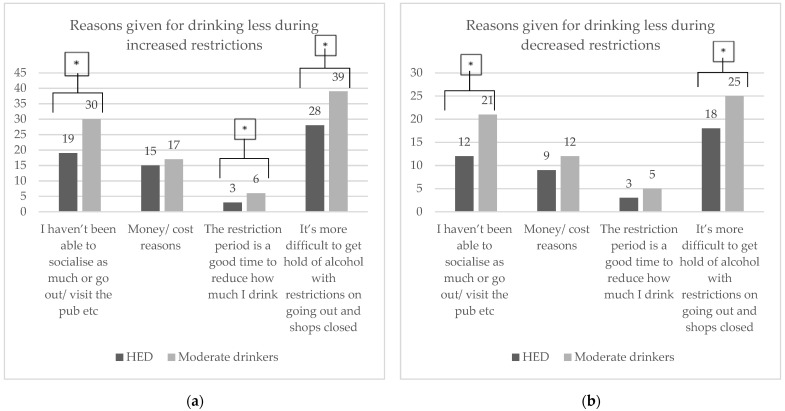
Reasons why HED and moderate drinkers reported drinking less alcohol during (**a**) increased lockdown restrictions and (**b**) decreased lockdown restrictions shown as percentages (values marked with an * are significantly different, *p* < 0.05).

**Table 1 ijerph-19-02422-t001:** Instrument domains, variables, and measures.

Domains	Variables	Measures
**Socio-demographics**	Date of birth	Numerical input (year of birth)
	Sex	Male
		Female
	Ethnicity	Black African
		Coloured
		White
		Asian/other
	Province of residence	Eastern Cape
		Free State
		Gauteng
		Kwazulu-Natal
		Mpumalanga
		Limpopo
		Northern Cape
		North West
		Western Cape
	Change in monthly household income during COVID-19	Decreased income
		No change
		Increased income
	Aged 70+, serious medical condition, or immunocompromised	Yes
		No
	COVID-19 pandemic restrictions at the time of completing the survey	^1^ Total restriction
		^2^ Very restricted
		^3^ Moderate restriction
		^4^ Limited restriction
		^5^ Social distancing only
		^6^ No restriction
** consumption, prevalence of HED, illegal alcohol purchasing and effect of alcohol on reducing drinking and social distancing of males and females that consume alcohol**	Frequency of alcohol consumption	≥1 times per day
		1–6 times a week
		1–4 times per month
		Less than once a month
		1–3 times per year
	Classified as heavy episodic drinkers *	Yes
		No
	Heavy episodic drinking frequency	Daily
		Weekly
		Monthly
		Less than monthly
	Illegal alcohol purchasing	Yes
		No
	First time buying alcohol illegally	Yes
		No
	Restrictions made it harder to cut down on drinking	Yes
		No
	Alcohol makes social/physical distancing more difficult	Yes
		No
**Pattern of alcohol** **consumption during** **increased and decreased levels of lockdown**	During the COVID-19 pandemic restrictions, did your drinking change from your usual level as the restrictions?	Drinking more
		Drinking the same
		Drinking less
**Behaviour change**	What are the reasons for drinking more during increased and decreased restrictions? You can tick more than one	It helps me relax/switch off I have been bored I have been feeling stressed out/anxious I have been spending more time drinking socially with people in my household I don’t need to get up for work/study More online social occasions (e.g., catching up online over a drink) People in my household have been encouraging me to drink more Celebration when COVID-19 restrictions are lifted/eased? I can go to pubs/bars now Alcohol shops are open now I can see my friends and drink with them now
	What are the reasons for drinking less during increased and decreased restrictions? You can tick more than one	I haven’t been able to socialise as much or go out/visit the pub, etc. Money/cost reasons The restriction period is a good time to reduce how much I drink It’s more difficult to get hold of alcohol with restrictions on going out and shops closed Physical health reasons (e.g., weight, health condition, to be healthier) Mental health reasons (e.g., anxiety, depression) I don’t like to drink around family/children while at home Family/relationship reasons I am living in an alcohol-free household Someone suggested I should reduce my drinking Work/study/sporting reasons I’ve been sick I have had COVID-19

^1^ Stay at home—cannot leave house (or only with permission for essentials). ^2^ Stay at home—can leave for food/medical/exercise. Non-essential businesses/schools are closed. Public gatherings are banned. Physical distancing is required. ^3^ Stay at home—can interact with a few people outside your household. Some businesses are open. School may/may not be open. Public gatherings are banned. Physical distancing is required. ^4^ Most businesses, schools, and workplaces are open; gatherings of people are allowed but size is restricted; physical distancing is required. ^5^ Physical distancing only required. ^6^ Life as normal. * Heavy episodic drinking is when a person consumes more than 6 drinks per occasion monthly or more frequently.

**Table 2 ijerph-19-02422-t002:** Socio-demographic information of males and females.

Variables	Total Dataset	Males	Females
	*n*(%) 95% CI *	*n*(%) 95% CI *	*n*(%) 95% CI *
**Age**			
18–34	105(13.5)11.2–16	33(14.2)10.1–19.1	72(13.2)10.5–16.2
35–44	141(18.1)15.5–20.9	28(16.3)12–21.5	103(18.9)15.8–22.3
45–54	165(21.2)18.4–24.2	41(17.6)13.1–22.9	124(22.7)19.3–26.4
55–64	228(29.3)26.2–32.5	64(27.5)22–33.5	164(30)26.3–34
≥65	140(18)15.4–20.8	57(24.5)19.3–30.3	83(15.2)12.4–18.4
**Ethnicity**			
Black African	73(9.2)7.4–11.4	25(10.5)7.1–14.9	48(8.7)6.7–11.2
Coloured	35(4.4)3.2–6	10(4.2)2.2–7.3	25(4.5)3–6.5
White	658(83.2)80.5–85.7	196(82.4)77.1–86.8	462(83.5)80.3–86.5
Asian/other	25(3.2)2.1–4.6	7(2.9)1.3–5.7	18(3.3)2–5
**Province**			
Eastern Cape	63(7.9)6.2–10	21(8.7)5.6–12.7	42(7.6)5.6–10
Free State	30(3.8)2.6–5.3	9(3.7)1.9–6.7	21(3.8)2.4–5.6
Gauteng	319(40.1)36.8–43.6	99(40.9)34.9–47.2	220(39.8)35.8–43.9
Kwazulu-Natal	120(15.1)12.7–17.7	31(12.8)9–17.5	89(16.1)13.2–19.3
Mpumalanga	31(3.9)2.7–5.4	10(4.1)2.1–7.2	21(3.8)2.4–5.6
Limpopo	27(3.4)2.3–4.8	9(3.7)1.9–6.7	18(3.3)2–5
Northern Cape	11(1.4)0.7–2.4	2(0.8)0.2–2.6	9(1.6)0.8–2.9
North West	22(2.8)1.8–4.1	8(3.3)1.6–6.1	14(2.5)1.5–4.1
Western Cape	172(21.6)18.9–24.6	53(21.9)17–27.4	119(21.5)18.2–25.1
**Change in monthly net household income since the COVID-19 pandemic restrictions**			
Decreased income	489(65.4)61.9–68.7	149(63.9)57.6–69.9	340(66)61.8–70
No change	220(29.4)26.2–32.8	66(28.3)22.8–34.4	154(29.9)26.1–34
Increased income	39(5.2)3.8–7	18(7.7)4.8–11.7	21(4.1)2.6–6.1
**Aged 70+, serious medical condition, or immunocompromised**			
Yes	97(12.6)10.4–15.1	27(11.6)8–16.2	70(13.1)10.4–16.1
No	672(87.4)84.9–89.6	206(88.4)83.8–92	466(86.9)83.9–89.6
**COVID-19 pandemic restrictions at the time of completing the survey**			
^1^ Total restriction	7(0.9)0.4–1.7	1(0.4)0–1.9	6(1.1)0.5–2.2
^2^ Very restricted	95(11.9)9.8–14.3	24(9.9)6.6–14.2	71(12.8)10.2–15.8
^3^ Moderate restriction	406(51)47.5–54.5	113(46.7)40.5–53	293(52.9)48.7–57
^4^ Limited restriction	248(31.2)28–34.4	89(36.8)30.9–43	159(28.7)25.1–32.6
^5^ Social distancing only	19(2.4)1.5–3.6	11(4.5)2.4–7.7	8(1.4)0.7–2.7
^6^ No restriction	21(2.6)1.7–3.9	4(1.7)0.6–3.9	17(3.1)1.9–4.8

^1^ Stay at home—cannot leave house (or only with permission for essentials). ^2^ Stay at home—can leave for food/medical/exercise. Non-essential businesses/schools are closed. Public gatherings are banned. Physical distancing is required. ^3^ Stay at home—can interact with a few people outside your household. Some businesses are open. School may/may not be open. Public gatherings are banned. Physical distancing is required. ^4^ Most businesses, schools, and workplaces are open; gatherings of people are allowed but size is restricted; physical distancing is required. ^5^ Physical distancing only required. ^6^ Life as normal. * Confidence intervals shown are for the proportions.

**Table 3 ijerph-19-02422-t003:** Patterns of alcohol consumption, HED, illegal alcohol purchasing, and influence of alcohol on cutting down drinking and social distancing of males and females who consume alcohol only.

Variables	Total Dataset	Males	Females
	*n*(%) 95% CI **	*n*(%) 95% CI **	*n*(%) 95% CI **
**Frequency of alcohol consumption**			
≥1 times per day	195(26.8)23.7–30.1	81(35.7)29.7–42.1	114(22.8)19.2–26.6
1–6 times a week	388(53.3)49.7–56.9	110(48.5)42–54.9	278(55.5)51.1–59.8
1–4 times per month	98(13.5)11.1–16.1	24(10.6)7.1–15.1	74(14.8)11.9–18.1
Less than once a month	37(5.1)3.7–6.9	10(4.4)2.3–7.7	27(5.4)3.7–7.6
1–3 times per year	10(1.4)0.7–2.4	2(0.9)0.2–2.8	8(1.6)0.8–3
**Classified as Heavy episodic drinkers ***			
Yes	346(48.5)44.9–52.2	137(61.4)54.9–67.6	209(42.7)38.3–47.1
No	367(51.5)47.8–55.1	86(38.6)32.4–45.1	281(57.3)52.9–61.7
**Heavy episodic drinking frequency**			
Daily	82(17.9)14.6–21.6	45(27.8)21.3–35	37(12.5)9.1–16.6
Weekly	182(39.7)35.3–44.3	68(42)34.6–49.7	114(38.5)33.1–44.1
Monthly	82(17.9)14.6–21.6	24(14.8)10–20.9	58(19.6)15.4–24.4
Less than monthly	112(24.5)20.7–28.5	25(15.4)10.5–21.6	87(29.4)24.4–34.8
**Illegal alcohol purchasing**			
Yes	438(55.3)51.8–58.7	146(60.8)54.6–66.8	292(52.9)48.7–57
No	354(44.7)41–48.2	94(39.2)33.2–45.4	260(47.1)43–51.3
**First time buying alcohol illegally**			
Yes	361(84)80.3–87	108(76.1)68.6–82.5)	253(87.8)83.7–91.2
No	69(16)12.8–19.7	34(23.9)17.5–31.4	35(12.2)8.8–16.3
**Restrictions made it harder to cut down on drinking**			
Yes	130(29.6)25.5–34	31(22.6)16.2–30.2	99(32.8)27.7–38.2
No	309(70.4)66–74.5	106(77.4)69.8–83.8	203(67.2)61.8–72.3
**Alcohol makes social/physical distancing more difficult**			
Yes	109(13.8)11.6–16.4	30(12.6)8.8–17.2	79(14.4)11.6–17.5
No	679(86.2)83.6–88.4	209(87.4)82.8–91.2	470(85.6)82.5–88.4

* Heavy episodic drinking is when a person consumes more than 6 drinks per occasion monthly or more frequently. ** Confidence intervals shown are for the proportions.

**Table 4 ijerph-19-02422-t004:** Socio-demographic information and drinking pattern of heavy episodic drinkers compared to moderate drinkers*.

Variables	Heavy Episodic Drinkers	Moderate Drinkers	*p*-Value
	*n*(%) 95% CI **	*n*(%) 95% CI **	(Chi-Square Test)
**Sex**			
Male	137(39.6)34.5–44.8	86(24.4)19.3–28	<0.001
Female	209(60.4)55.2–65.5	281(76.6)72–80.7	
**Age**			
18–34	57(17)13.2–21.3	43(11.7)8.8–15.3	<0.001
35–44	80(23.8)19.5–28.6	45(12.3)9.2–16	
45–54	83(24.7)20.3–29.5	66(18)14.4–22.2	
55–64	79(23.5)19.2–28.3	132(36.1)31.3–41.1	
>65	37(11)8–14.7	80(21.9)17.9–26.3	
**Ethnicity**			
Black African	37(10.7) 7.8–14.3	26(7)4.7–9.9	0.343
Coloured	15(4.3) 2.6–6.9	14(3.8)2.2–6.1	
White	282(81.5) 77.2–85.3	319(85.8)81.9–89.0	
Asian/Other	12(3.5) 1.9–5.8	13(3.5)2.0–5.7	
**Province**			
Eastern Cape	26(7.4) 5.0–10.6	28(7.5) 5.1–10.5	0.956
Free State	11(3.2) 1.7–5.4	17(4.5) 2.8–7.0	
Gauteng	146(41.8)36.7–47.1	149(39.8)35–44.9	
Kwazulu-Natal	52(14.9)11.5–18.9	59(15.8)12.4–19.7	
Mpumalanga	16(4.6)2.8–7.2	12(3.2)1.8–5.4	
Limpopo	11(3.2)1.7–5.4	12(3.2)1.8–5.4	
Northern Cape	4(1.1)0.4–2.7	6(1.6)0.7–3.3	
North West	8(2.3)1.1–4.3	11(2.9)1.6–5	
Western Cape	75(21.5)17.4–26	80(21.4)17.5–25.8	
**Change in monthly net household income since the COVID-19 pandemic restrictions**			
Decreased income	240(70.2)65.2–74.8	222(59.8)54.8–64.7	0.004
No change	82(24)19.7–28.7	131(35.3)30.6–40.3	
Increased income	20(5.8)3.7–8.7	18(4.9)3–7.4	
**Aged 70+, serious medical condition, or immunocompromised**			
Yes	31(9.1)6.4–12.5	60(16.7)13.1–20.8	0.003
No	309(90.9)87.5–93.6	300(83.3)79.2–86.9	
**COVID-19 pandemic restrictions**			
^1^ Total restriction	3(0.8)0.2–2.2	4(1)0.3–2.4	0.150
^2^ Very restricted	43(11.8)8.8–15.4	47(12)9.1–15.5	
^3^ Moderate restriction	182(49.9)44.8–55	199(50.9)45.9–55.8	
^4^ Limited restriction	112(30.7)26.1–35.6	131(33.5)29–38.3	
^5^ Social distancing only	13(3.6)2–5.8	5(1.3)0.5–2.8	
^6^ No restriction	12(3.3)1.8–5.5	5(1.3)0.5–2.8	
**Frequency of alcohol consumption**			
≥1 times per day	121(33.2)28.5–38.1	78(20.1)16.3–24.3	<0.001#
1–6 times a week	209(57.3)52.1–62.3	188(48.5)43.5–53.4	
1–4 times per month	25(6.8)11.7–16.7	81(20.9)17.1–25.1	
Less than once a month	10(2.7)4.1–7.4	32(8.2)5.8–11.3	
1–3 times per year	0	9(2.3)1.2–4.2	
**Illegal alcohol purchasing**			
Yes	278(76.2)71.6–80.3	171(44.2)39.3–49.2	<0.001
No	87(23.8)19.7–28.4	216(55.8)50.8–60.7	
**First time buying alcohol illegally**			
Yes	220(80.9)75.9–85.2	153(90)84.8–93.8	0.010
No	52(19.1)14.8–24.1	17(10)6.2–15.2	
**Restrictions made it harder to cut down on drinking**			
Yes	92(38)32.1–44.2	35(20)14.6–26.4	<0.001
No	150(62)55.8–67.9	140(80)73.6–85.4	
**Alcohol makes social/physical distancing more difficult**			
Yes	38(10.9)7.9–14.5	64(17)13.4–21	0.018
No	311(89.1)85.5–92.1	313(83)79–86.6	

* Based on question 165 ‘How often did you have 6 or more drinks of alcohol on a single occasion during the COVID-19 pandemic restrictions?’; anything more regular than monthly (including monthly) was classified as HED while anything less than monthly was classified as moderate. ^1^ Stay at home—cannot leave house (or only with permission for essentials). ^2^ Stay at home—can leave for food/medical/exercise. Non-essential businesses/schools are closed. Public gatherings are banned. Physical distancing is required. ^3^ Stay at home—can interact with a few people outside your household. Some businesses are open. School may/may not be open. Public gatherings are banned. Physical distancing is required. ^4^ Most businesses, schools, and workplaces are open; gatherings of people are allowed but size is restricted; physical distancing is required. ^5^ Physical distancing only required. ^6^ Life as normal. #Fisher’s exact test. ** Confidence intervals shown are for the proportions.

**Table 5 ijerph-19-02422-t005:** Multiple logistic regression of socio-demographic and alcohol intake variables with heavy episodic drinking.

Variables	AOR *	95%CI **	*p*-Value
**Sex**			
Male	(ref)		
Female	0.30	0.13–0.70	0.006
**Age**			
18–34	(ref)		
35–44	1.58	0.50–4.99	0.439
45–54	1.17	0.39–3.54	0.775
55–64	0.46	0.16–1.31	0.144
≥65	0.23	0.06–0.91	0.037
**Change in monthly net household income since the COVID-19 pandemic restrictions**			
Decreased income	(ref)		
No change	0.97	0.42–2.22	0.937
Increased income	1.46	0.34–6.19	0.607
**Aged 70+, serious medical condition, or immunocompromised**			
Yes	(ref)		
No	1.21	0.39–3.79	0.738
**Frequency of alcohol consumption**			
≥1 times per day	(ref)		
1–6 times a week	0.74	0.33–1.66	0.464
1–4 times per month	0.13	0.03–0.45	0.001
Less than once a month	0.56	0.10–3.07	0.507
**Illegal alcohol purchasing**			
Yes	(ref)		
No	0.20	0.10–0.40	<0.001
**First time buying alcohol illegally**			
Yes	(ref)		
No	1.72	0.63–4.66	0.287
**Restrictions made it harder to cut down on drinking**			
Yes	(ref)		
No	0.49	0.24–1.00	0.051
**Alcohol makes social/physical distancing more difficult**			
Yes	(ref)		
No	1.68	0.60–4.66	0.321

*Adjusted odds ratio (AOR). ** Confidence intervals shown are for the proportions.

## Data Availability

The datasets analysed during the current study are available from the corresponding author on reasonable request.

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
