# Peer review of "Did COVID-19-Related Alcohol Sales Restrictions Reduce Alcohol Consumption? Findings from a National Online Survey in South Africa"

_ijerph, 2022, doi:10.3390/ijerph19042422_

Round 1

Reviewer 1 Report

The paper focuses on an interesting theme: how COVID-19 related alcohol sales restrictions affected alcohol consumption in South Africa.  Data were collected using a self-reported Facebook survey.

The theme is well founded through a consistent literature review. The objectives are correctly defined. The methodology is well-designed and is consistent with the objectives of the study. The interpretation and discussion of results is clear, objective, and consistent. The conclusions summarize well the results obtained and are consistent with the work presented. However, there are some details that can be improved:

  • p1 (lines 21-22), p.4 (line 143, 147,148) – replace “Fishers exact test” with “Fisher's exact test”
  • p1 (lines 22), p.4 (line 144) – replace “Students T-test” with “Student’s T-test”
  • p1 (lines 24) - integrate “n=798 (68.4% female)”into a sentence
  • In the description of the methodology, the authors refer to the application of a questionnaire through Facebook. Slovin's Formula was used to determine the sample size (lines 83-103). The Slovin’s Formula assumes that the sample is taken at random. Since the sample in question is obtained for convenience, it makes no sense to apply the Sloven formula. It should only be mentioned that a convenience sample was used.
  • p3 (lines 104-137) - The description of the questionnaire is a little confusing. It is suggested to organize this information in a table with the questions and respective answer categories.
  • p4 (line 166) - The content of the information presented in Table 1 must be clarified in the legend, namely about the confidence intervals. Are these confidence intervals for the proportions shown?
  • p6 (line 190) - The content of the information presented in Table 2 must be clarified in the legend, namely about the confidence intervals. Are these confidence intervals for the proportions shown?
  • p7 (line 208) - The content of the information presented in Table 3 must be clarified in the legend, namely about the confidence intervals. Are these confidence intervals for the proportions shown?
  • pp9-11 - Figures 1, 2, 3 and 4 should have similar formatting. The y-axis legend is different, the bar format is also different. The legend in Figure 2 must be carefully revised.
  • p11 (lines 262-273) - The authors present some results of a multinomial logistic regression. It is, however, necessary to explain the adjusted model in more detail (which variables entered the model, regression method used, goodness of fit, etc.)
  • p13 (line 304) – Replace [24] (above the line) with [24]

Author Response

Section or Page number

Comments

Reviewer one

p1 (lines 21-22), p.4 (line 143, 147,148) – replace “Fishers exact test” with “Fisher's exact test”

Thank you for noting these mistakes. The writing of Fisher’s exact test has been corrected.

p1 (lines 22), p.4 (line 144) – replace “Students T-test” with “Student’s T-test”

Thank you for noting these mistakes. The writing of Student’s T-test has been corrected.

p1 (lines 24) - integrate “n=798 (68.4% female)”into a sentence

Integrated wording into a sentence.

In the description of the methodology, the authors refer to the application of a questionnaire through Facebook. Slovin's Formula was used to determine the sample size (lines 83-103). The Slovin’s Formula assumes that the sample is taken at random. Since the sample in question is obtained for convenience, it makes no sense to apply the Sloven formula. It should only be mentioned that a convenience sample was used.

Thank you for this comment. We agree that adding the explanation that it was indeed a convenience sample will clarify the methodology. We have removed reference to Slovin’s formula as suggested.

We have updated the referencing and numbering after removing the reference to Slovin’s formula.

p3 (lines 104-137) - The description of the questionnaire is a little confusing. It is suggested to organize this information in a table with the questions and respective answer categories.

Thank you for this comment. Please find that we have now included a table with the questions and response categories in the methods section.

p4 (line 166) - The content of the information presented in Table 1 must be clarified in the legend, namely about the confidence intervals. Are these confidence intervals for the proportions shown?

Added the wording in the legend of the table: * Confidence intervals shown are for the proportions

p6 (line 190) - The content of the information presented in Table 2 must be clarified in the legend, namely about the confidence intervals. Are these confidence intervals for the proportions shown?

Added the wording in the legend of the table: * Confidence intervals shown are for the proportions

p7 (line 208) - The content of the information presented in Table 3 must be clarified in the legend, namely about the confidence intervals. Are these confidence intervals for the proportions shown?

Added the wording in the legend of the table: * Confidence intervals shown are for the proportions

p9-11 - Figures 1, 2, 3 and 4 should have similar formatting. The y-axis legend is different, the bar format is also different. The legend in Figure 2 must be carefully revised.

Thank you for this comment. Please find that all the figures have now been formatted to look the same and we have improved the legends and headings.

p11 (lines 262-273) - The authors present some results of a multinomial logistic regression. It is, however, necessary to explain the adjusted model in more detail (which variables entered the model, regression method used, goodness of fit, etc.)

Please find that we have included much more information on how the logistic regression was done under the heading of statistics page 5.

After identifying variables that were significantly associated with the dependent binary variable of HED and Moderate drinker they were entered to compile a standard multiple logistic regression in STATA. The variables that were entered into the model to calculate the adjusted odds ratio (AOR) were the following: Sex, Age, Change in monthly net household income since the COVID-19 pandemic restrictions, Aged 70+, serious medical condition or immunocompromised, Frequency of alcohol consumption, Illegal alcohol purchasing, First time to buy alcohol illegally, Restrictions made it harder to cut down on drinking, Alcohol makes social/physical distancing more difficult. Multicollinearity was assessed by examining correlations between predictors. No two predictors had a correlation of more than 0.5. Model fit was checked using an adaptation of Hosmer Lemeshow’s Goodness of Fit Test, and all models indicated appropriate fit. P-values less than 0.05 were considered statistically significant.

p13 (line 304) – Replace [24] (above the line) with [24]

Corrected the superscript to normal text.

Reviewer 2 Report

I consider that the submitted manuscript whose main objective is to investigate the sociodemographic profile and its alcohol consumption behavior changes during the different levels of confinement restrictions and factors related to HED during the first months of the COVID-19 pandemic in South Africa, is a suitable and interesting subject, to be published in the special issue of the magazine.

However, there are certain modifications or suggestions that I make below and that the authors must respond to in order to increase its quality and comply with the criteria and instructions for publication, set out by the journal itself. These are:

Summary:

In order to comply with the regulations of the journal, authors are urged to eliminate the numbers placed in parentheses from the abstract, prior to the presentation of each section of the abstract.

Keywords:

Three to ten relevant keywords should be added after the abstract. It is recommended that these be specific to the article, but reasonably common within the discipline. The authors put "anxiety and depression" This would be two. Before the last word, they must put and. Authors are encouraged to review the rest of the keywords to make them more specific and clear.

Introduction:

 Regarding alcohol consumption restrictions;

  1. These restrictions were at the beginning of the pandemic, were they later…? The authors should state which were the periods of the restrictions regarding alcohol consumption in SA and not only mention that it was for three months.
  2. What agency or managing body of the Country imposed these restrictions? Is there a law that makes reference to these restrictions in relation to alcohol consumption? All this information, it would be important to mention it in the text.

Methodology:

The authors must indicate in the methodology section the ethical conditions in which the research is carried out (Informed consent signature, voluntary participation, information on data protection ... ..)

Could you answer these questions? How did you ensure that the participants were over 18 years of age and resident in SA?

Although the data collection procedure was carried out through a self-administered survey, the authors should indicate the type of sampling (Intentional, snowball ...)

The survey is a validated or ad hoc survey, conducted for the purpose of the study (this must be indicated in the text)

Authors are recommended to write the Methodology section in a more structured way for an easier understanding of the text and separate it into ordered sections (type of study, population, sample, study variables, instruments, procedure and ethical considerations ).

The authors must indicate in the methodology section the ethical conditions in which the research is carried out (Informed consent signature, voluntary participation, information on data protection ...) not only say that the study is approved by the university .

Conclusions:

It is recommended that the authors present the conclusions in a more summarized and direct way without repeating what is said in the discussion of the work.

Bibliographic references:

Authors are requested to review the bibliographic references section and modify all those that do not comply with the journal's regulations (Due to missing elements, not putting the name of the journal abbreviated or not putting the year of publication in print Bold, for example, in refs. 1, 2, 3, 4, 7, 8, 9, 19, 22, 23, 24, among others. You can find the rules regarding the writing of bibliographic references at the link: https : //www.mdpi.com/journal/ijerph/instructions

Figures:

The titles of the figures should be written all in bold or all in normal type, so that there are no differences in the style of the manuscript

Tables:

The authors must adapt the size of tables 1 and 4 since there are results of variables that are displaced from the corresponding place.

Author Response

Reviewer Two

Summary:

In order to comply with the regulations of the journal, authors are urged to eliminate the numbers placed in parentheses from the abstract, prior to the presentation of each section of the abstract.

Thank you for bringing this to our attention. Numbers have been removed.

Keywords:

Three to ten relevant keywords should be added after the abstract. It is recommended that these be specific to the article, but reasonably common within the discipline. The authors put "anxiety and depression" This would be two. Before the last word, they must put and. Authors are encouraged to review the rest of the keywords to make them more specific and clear.

Thank you for this comment. We have separated the words anxiety and depression. We have added the word ‘and’ before the last word. We have added the term ‘alcohol policies’.

Introduction:

 Regarding alcohol consumption restrictions;

  1. These restrictions were at the beginning of the pandemic, were they later…? The authors should state which were the periods of the restrictions regarding alcohol consumption in SA and not only mention that it was for three months.
  2. What agency or managing body of the Country imposed these restrictions? Is there a law that makes reference to these restrictions in relation to alcohol consumption? All this information, it would be important to mention it in the text.

1.      Thank you for this comment, we have now included the dates and timing of the alcohol sales restrictions to date.

2.      We added the words ‘the declaration of a national state of disaster by the government’. To clarify that these restrictions were mandated by laws and the words ‘were instituted by amending the disaster management act’ were included and the laws are referenced as references 4 and 7.

Methodology:

The authors must indicate in the methodology section the ethical conditions in which the research is carried out (Informed consent signature, voluntary participation, information on data protection ... ..)

Could you answer these questions? How did you ensure that the participants were over 18 years of age and resident in SA?

Although the data collection procedure was carried out through a self-administered survey, the authors should indicate the type of sampling (Intentional, snowball ...)

The survey is a validated or ad hoc survey, conducted for the purpose of the study (this must be indicated in the text)

Authors are recommended to write the Methodology section in a more structured way for an easier understanding of the text and separate it into ordered sections (type of study, population, sample, study variables, instruments, procedure and ethical considerations ).

The authors must indicate in the methodology section the ethical conditions in which the research is carried out (Informed consent signature, voluntary participation, information on data protection ...) not only say that the study is approved by the university .

Thank you for these important comments. We have now included sub-headings to the methodology section and included the following information:

Type of study: Online ad-hoc self-reported survey.

Ethical considerations:

Participants were anonymized and responses were stored in secure firewalled facilities that are password protected. Access was only given to research team members. Information was given upfront regarding alcohol addiction help lines and contact details of researchers for more information. Participants were assured that participation was voluntary and asked to indicate their agreement to participate by choosing ‘agree’ or ‘not agree’ to participate, which were used to indicate signatory agreement of the informed consent sheets. This research was given ethical approval by the South African Medical Research Council (ref: EC017-6/2020) and the University of the Western Cape Biomedical Research Ethics Council (ref: BM21/5/11).

More information was included for the population and sample stating that is was a convenience sample.

More information on the instrument and procedures were included.

Conclusions:

It is recommended that the authors present the conclusions in a more summarized and direct way without repeating what is said in the discussion of the work.

We have reduced the conclusion and removed repetitive information and references.

Bibliographic references:

Authors are requested to review the bibliographic references section and modify all those that do not comply with the journal's regulations (Due to missing elements, not putting the name of the journal abbreviated or not putting the year of publication in print Bold, for example, in refs. 1, 2, 3, 4, 7, 8, 9, 19, 22, 23, 24, among others. You can find the rules regarding the writing of bibliographic references at the link: https : //www.mdpi.com/journal/ijerph/instructions

Please note that not all the references are from journal publications. Some are opening remarks, data repositories, legal acts or regulations. Please find the corrections made and types of references for each reference included below.

1.      Opening remarks

2.      Data repository

3.      Declaration of state of disaster

4.      Legal document Act

5.      Abbreviated the journal name

6.      Abbreviated the journal name

7.      Legal document Act

8.      Website (added website address)

9.      Report

10.  Abbreviated the journal name

11.  Statistics report

12.  Abbreviated journal name

13.  Added volume, issue and page numbers.

14.  Abbreviated journal name

15.  Abbreviated journal name

16.  Website

17.  Abbreviated journal name

18.  Abbreviated journal name

19.  Website

20.  Software

21.  Software

22.  Software

23.  Abbreviated journal name

24.  Abbreviated journal name

25.  Abbreviated journal name

26.  Correct

27.  Abbreviated journal name

28.  Working paper

29.  Abbreviated journal name

30.  Report

Figures:

The titles of the figures should be written all in bold or all in normal type, so that there are no differences in the style of the manuscript

Removed bold from heading of Figure 1a and b and made all headings look the same.

Also in Figure 2 heading.

Tables:

The authors must adapt the size of tables 1 and 4 since there are results of variables that are displaced from the corresponding place.

Please find tables have been adapted.

Reviewer 3 Report

I am grateful for the opportunity to review the manuscript presented to me. I hope that the comments in the review would be helpful. I believe the paper is worth considering for publication, however requires minor revision.

Author Response

Reviewer Three

Introduction:

Remove repetition

Repetition of WHO abbreviation explanation removed.

Methods

please divide into sections, eg Participants, statistical analysis etc .. The methodology will be clearer.

This has been implemented. Thank you for this valuable comment.

has the tool been validated? if so, please describe

This tool has not been validated. We have now clarified this in the methodology, stating that the tool was an ad-hoc tool created for this survey.

was the tool published? if so, please provide citation.

This tool was not published.

Table 3

in my opinion, P-value cannot equal zero. If the severity is very high, it is necessary to write: p <0.001. Please improve throughout the work (tables and description)

Thank you for this comment. We have now changed the p-values to be <0.001 when they state 0.000.

Discussion

P13 correct superscript 24

Done.

P13Use upper case States

Thank you for noticing this. Change made.

P13 I understand the concept of work but it's hard to call a comparison "pre" and "post" a pandemic because it's not over yet.

Thank you for this comment. We have changed the wording to be “pre- and continuing pandemic....”

P14 Most of the people (with the age division applied) are middle-aged and older people. In addition, the survey was carried out via the Internet, which at the beginning determines people who at this age can use it enough to fill in the questionnaire. I think that the described problem at work is much deeper - what about people who do not have the Internet or do not know how to use it? An additional limitation of the study is the inequality of one of the groups (55-64), because the results would be completely different, e.g. if the largest group were young people.

We agree with this statement. Please find that we have added that the older age range was a limitation. Please find specific reference to the internet accessibility in lines 491-500.

Conclusions

please specify the conclusions max. 3-4 sentences and do not use citations in this part.

Thank you for this comment. We have adapted the conclusion to be much more succinct and removed the citations.

Round 2

Reviewer 2 Report

Dear authors, congratulate you for the work done after the first round of revisions.

Thank you very much for responding so effectively to the suggested changes regarding:

Compliance with the regulations of the journal in the Abstract, keywords and bibliographic references sections

  • The justification requested regarding the regulations that restricted the consumption of alcohol
  • The inclusion of ethical considerations
  • Inclusion and expansion in relation to the data collection process
  • The reduction of at the conclusion of the work
  • The adaptation of those elements indicated in the tables of the study

However, and as minor modifications in this second revision of the article, I urge the authors to make the following modifications:

  1. Ethical considerations should be placed at the end of the methodology and not after the type of study
  2. Work on the modification of table Nº1 and put it in APA Regulations. Also adjust it so that the domain column does not cut the words

Greetings
